# Determination of Levamisole and Mebendazole and Its Two Metabolite Residues in Three Poultry Species by HPLC-MS/MS

**DOI:** 10.3390/foods10112841

**Published:** 2021-11-17

**Authors:** Pengfei Gao, Peiyang Zhang, Yawen Guo, Zhaoyuan He, Yuhao Dong, Yayun Tang, Fanxun Guan, Tao Zhang, Kaizhou Xie

**Affiliations:** 1College of Animal Science and Technology, Yangzhou University, Yangzhou 225009, China; MZ120201383@yzu.edu.cn (P.G.); zhangpy0218@163.com (P.Z.); DX120200135@yzu.edu.cn (Y.G.); MZ120191027@yzu.edu.cn (Z.H.); tyy199727@163.com (Y.T.); dahai19981118@163.com (F.G.); zhangt@yzu.edu.cn (T.Z.); 2Joint International Research Laboratory of Agriculture & Agri-Product Safety, Yangzhou University, Yangzhou 225009, China; 3College of Veterinary Medicine, Nanjing Agricultural University, Nanjing 210095, China; dongyuhao@njau.edu.cn

**Keywords:** chicken meat, goose meat, duck meat, veterinary drugs, validation method

## Abstract

A high-performance liquid chromatography-tandem mass spectrometry (HPLC-MS/MS) method was developed to simultaneously analyze levamisole (LMS) and mebendazole (MBZ) and its two metabolites, 5-hydroxymebendazole (HMBZ) and 2-amino-5-benzoylbenzimidazole (AMBZ), in poultry muscle (chicken, duck and goose). In the sample preparation process, basic ethyl acetate was used as the extraction agent, and the extracted samples were back-extracted with hydrochloric acid, purified by Oasis MCX solid-phase extraction (SPE) cartridges, and reconstituted in the initial mobile phase after being blown dry with nitrogen. Chromatographic separation was performed on an Xbridge C_18_ column (4.6 mm × 150 mm, 5 μm) with 0.1% formic acid in water and acetonitrile as the mobile phases, and gradient elution was performed at a flow rate of 0.6 mL/min and a column temperature of 35 °C. In blank poultry muscle samples, the spiked concentrations of LMS, MBZ, HMBZ, and AMBZ were within the range of the limit of quantitation (LOQ) to 25 μg/kg. The peak areas of the four target drugs had a good linear relationship with the concentration, and the determination coefficient (R^2^) values were higher than 0.9990. The average recoveries of LMS, MBZ, HMBZ, and AMBZ were 86.77–96.94%; the intraday relative standard deviations (RSDs) were 1.75–4.99% at LOQ, 0.5 maximum residue limit (MRL), 1.0 MRL, and 2.0 MRL; the interday RSDs were 2.54–5.52%; and the LODs and LOQs were 0.04–0.30 μg/kg and 0.12–0.80 μg/kg, respectively.

## 1. Introduction

The development of intensive livestock and poultry farming has increased the spread of parasitic diseases, causing great economic losses to farmers. Levamisole (LMS) and mebendazole (MBZ) are widely used in the livestock and poultry industries as efficient, broad-spectrum insect repellents. LMS blocks the reduction of fumaric acid to succinic acid by inhibiting the activity of fumarase in the parasite, thus affecting sugar metabolism and reducing adenosine triphosphate (ATP), leading to parasite paralysis and death [1]. LMS is also an effective immunostimulatory substance in the regulation of phagocytes and macrophage activation factors and antibody responses [2]. MBZ is a benzimidazole drug, and its main mechanism of action is to inhibit the absorption and utilization of glucose by parasites, hinder the production of ATP or extend the half-life of cell hydrolase, and accelerate cell lysis, thereby leading to glycogen depletion in the parasite and preventing parasite survival and reproduction [3]. Since LMS and MBZ have different mechanisms of action against insects, combining the two can provide a synergistic effect, allow dosage reductions, delay the generation of drug resistance, reduce the toxicity and side effects of the drugs, and improve the treatment effects of the single drugs as well as the immune synergistic effect of LMS [4]. In mouse experiments, Bennet et al. used a combination of LMS and MBZ to prevent and treat the larval stage of *Mesocestoides corti* and showed that LMS as an adjuvant could enhance the immune response of the host [5]. When combined with the known deworming effect of MBZ, this enhanced immune response improved the efficacy of the benzimidazole derivative [6]. Thus, LMS and MBZ are used in many countries to treat different parasitic infections.

In general, residual drug levels can be controlled within a relatively safe range without causing harm to human health if the correct dosing method is followed and the drug withdrawal period is observed during livestock and poultry raising. The European Union [7], the United States [8] and South Korea [9] have set maximum residue limits (MRLs) for veterinary drugs and their metabolites in animal-origin foods. The MRL for LMS in poultry muscle is 10 μg/kg in the European Union [7] and the United States [8], and the MRL for MBZ and its two metabolites, 5-hydroxymebendazole (HMBZ) and 2-amino-5-benzoylbenzimidazole (AMBZ), is 60 μg/kg in South Korea [9]. However, South Korea has no regulations on the MRL of LMS in poultry tissues. The European Union and the United States do not have regulations on the MRLs of MBZ and its two metabolites in poultry tissues, although the European Union has corresponding regulations for sheep and horses. However, to increase economic benefits, some breeders fail to follow the prescribed medication regimen and the withdrawal period during poultry growth, resulting in residual drug levels in food exceeding the MRL. Additionally, excessive LMS enrichment in the human body can cause serious damage, such as cutaneous necrotizing vasculitis, granulocyte hypoxia, or effects on the nervous system [10]. MBZ, HMBZ and AMBZ have embryotoxic and teratogenic properties due to inhibition of tubulin and mitosis. Veterinary drug residues are an important global food safety concern, and to monitor pharmaceutical residues, particularly in poultry foods, there is a need to develop a universal and rapid analytical method that sensitively and accurately detects the amount of veterinary drug residue by simple sample preparation.

At present, the main detection methods for LMS and MBZ are immunoassays, gas chromatography (GC) and liquid chromatography (LC). LC methods mainly include high-performance liquid chromatography (HPLC), ultra-performance liquid chromatography (UPLC), and liquid chromatography-tandem mass spectrometry (LC-MS/MS). Guo et al. developed a colloidal gold immunochromatographic assay based on universal monoclonal antibodies for the simultaneous detection of benzimidazole drug residues in milk samples [11]. Although some studies have used GC for the analysis of LMS [12] and MBZ and its two metabolites [13], GC is not as widely used as LC or LC-MS/MS due to its basic properties and low volatility of these drugs. Fluorescence detection is ideal for fluorescence sensitivity and selectivity, but LMS and MBZ do not exhibit fluorescence and thus must be derivatized prior to analysis. Ultraviolet detection has the same applicability as fluorescence detection, and therefore, LC detection of LMS and MBZ and their metabolites in animal-derived food has mainly been performed with ultraviolet detection [14] and diode-array detection [15,16]. Mass spectrometry has the advantages of high recovery, high selectivity and good repeatability, so it can provide accurate relative molecular masses, extensive fragment structural information, greater qualitative stability, and higher detection efficiency for veterinary drug residues in animal foods. In recent years, there have been an increasing number of studies on HPLC-MS/MS detection of LMS or MBZ and its metabolite residues in animal-derived foods [17,18,19,20], but simultaneous detection methods for these drugs are rarely reported, and the primary matrices have been aquatic products [21], beef [22], pork [23] and milk [24]. Related research on other poultry muscles has not been reported.

Thus, we developed an HPLC-MS/MS method for the simultaneous determination of LMS, MBZ, HMBZ, and AMBZ residues in the muscle of poultry (chicken, duck and goose). The effects of different extractants and solid-phase extraction (SPE) cartridges on recovery were studied in detail, as well as the effects of the main HPLC-MS/MS conditions on other experimental results. The purpose of this study was to provide a new research method for the simultaneous detection of deworming drugs in livestock and poultry and improve the detection technology of LMS, MBZ, HMBZ, and AMBZ residues in poultry muscle. This study is of great significance for the promotion of animal food safety testing, human health and the safe development of animal husbandry and provides technical support and a scientific basis for the monitoring of veterinary drug residues.

## 2. Materials and Methods

### 2.1. Chemicals and Reagents

An analytical standard of LMS hydrochloride (purity ≥ 98%) was purchased from Beijing Solarbio Science & Technology Co., Ltd. (Beijing, China). An MBZ (purity ≥ 98%) standard was obtained from Shanghai Yuanye Bio-Technology Co., Ltd. (Shanghai, China). Hydroxybenzimidazole (96.28% purity) and amino MBZ (97.70% purity) were purchased from Labor Dr. Ehrenstorfer-Schafers (Augsburg, Germany). HPLC-grade methanol and acetonitrile were obtained from Tedia Company Inc. (Fairfield, OH, USA). Analytical-grade ethyl acetate, ammonium formate, formic acid, ammonia, sodium sulfate and KOH were supplied by Sinopharm Chemical Reagent Co. (Shanghai, China). Experimental ultrapure water was prepared by a Thermo Scientific Smart2-Pure ultrapure water preparation system (Thermo Fisher Corp., Waltham, MA, USA), and the resistance of the instrument was 18.2 MΩ/cm (25 °C). Organic nylon syringe filters (13 mm × 0.45 μm) were purchased from Anpel Laboratory Technologies (Shanghai) Inc. (Shanghai, China).

### 2.2. Preparation of Solution

Appropriate amounts of LMS, MBZ, HMBZ, and AMBZ standards converted to 100% by mass were weighed, placed in 10-mL brown volumetric flasks, dissolved in 1 mL of methanol and brought to volume to 10 mL with methanol. Standard stock solutions of LMS, MBZ, HMBZ, and AMBZ at 1.00 mg/mL were prepared and stored in a freezer at −70 °C for 2 months.

A 1.0 mg/mL standard stock solution was obtained from a −34 °C refrigerator, and 100 μg/mL, 10 μg/mL, 5 μg/mL, 1 μg/mL, 100 ng/mL and 10 ng/mL standard working solutions were prepared by a serial dilution method with 10:90 acetonitrile/0.1% formic acid aqueous solution (*v/v*). The standard working solution was stored stably for one month in a freezer at −34 °C.

The 5 μg/mL standard working solutions of LMS, MBZ, HMBZ, and AMBZ were diluted stepwise with 50% methanol aqueous solution containing 0.1% formic acid to prepare a 50 ng/mL mass spectral tuning solution.

Finally, 0.1 mol/L hydrochloric acid solution, 5% ammoniated methanol and 50% potassium hydroxide solution were accurately prepared as extraction solutions.

Using a graduated cylinder, 999 mL of ultrapure water was placed into a 1000 mL screw-type flask, and 1 mL of formic acid was added to prepare a 0.1% formic acid aqueous solution, which was used as one of the mobile phases after uniform mixing and degassing.

### 2.3. HPLC-MS/MS Instruments and Conditions

HPLC analysis was carried out on a Waters Alliance e2695 system (Waters Corp., Milford, MA, USA) equipped with an AB SCIEX Triple Quad™ 5500 triple quadrupole mass spectrometer (AB SCIEX Corp., Framingham, MA, USA) and was controlled by Analyst software (AB SCIEX Corp., Framingham, MA, USA). Chromatography was carried out on a Waters XBridgeTMC_18_ column (4.6 mm × 150 mm, 5 μm) at 35 °C. Mobile phase A was HPLC-grade acetonitrile, and mobile phase B was 0.1% formic acid water. Gradient elution was carried out at a flow rate of 0.6 mL/min as follows: 0–1 min, 90% B; 2–8 min, 52% B; and 9–10 min, 90% B.

The prepared spectral tuning solutions of LMS, MBZ, HMBZ, and AMBZ (50 ng/mL) standards were injected in constant current mode by a mass spectrometer needle pump, and the positive electrospray ionization (ESI+) scanning mode was selected. First, a Q1 scan was used with ESI+, and the collection time was set to 5 min. The scan rate was 200 Da/s, and the scan range was 100 MW (molecular weight of the compound to be optimized) +30 Da. The needle pump was operated at a flow rate of 10 µL/min. After stabilization, data were collected, and the abscissa corresponding to the peak center of the target compound was recorded as the precursor ion of the compound to be tested, that is, the mass-to-charge ratio (*m/z*) of the precursor ion. Then, in the product ion scanning mode, the product ion mass-to-charge ratio of each analyte precursor ion was determined within the range of the accurate mass-to-charge ratio of 50-precursor ion +30 Da, the initial value of collision energy (CE) was 5 eV, the CE value was manually adjusted (increased by 5 eV each time), the scanning rate was 200 Da/s, and the collection time was 5 min. The signal strength of the precursor ion was preferably 1/3 or 1/4 of the strongest fragment ion signal in the chromatogram; two product ions were selected as the qualitative ions, and the product ion with the strongest signal was the quantitative ion. Finally, the selected precursor ion and two product ions of the target analytes were combined into multiple reaction monitoring (MRM) ion pairs, the analysis time of each ion pair was reasonably allocated, and the CE and declustering potential of each ion pair were further optimized. The parameters were saved to preliminarily establish the MRM method.

The mass spectrometer was operated in the ESI+ scanning and MRM modes to monitor the most abundant precursor ions to determine the optimal fragment ion transitions for each analyte. The ESI voltage was optimized to 5500 V, and the ion source temperature was set to 550 °C. The atmospheric pressures of the curtain gas, collision gas, ion source spray gas, and auxiliary heating gas (nitrogen) were set to 35 psi, 8 psi, 50 psi and 5 psi, respectively. The collision chamber outlet voltage and intake voltage were set to 12 V and 10 V, respectively. The optimal settings for the CE and the deblocking voltage, which differed for each analyte to obtain the best molecular ion fragmentation, are presented in Table 1 for LMS, MBZ, HMBZ, and AMBZ, including the optimized conditions and retention times.

### 2.4. Preparation of Sample

The experiments in this study were authorized by the Ethics Committee of Yangzhou University and Jiangsu Jinghai Poultry Industry Group Co., Ltd. (Haimen, China) and were conducted in strict accordance with the recommendations of the Guide to the Protection and Use of Laboratory Animals in Jiangsu Province. Seventy-day-old Haiyang yellow chickens (Jiangsu Jinghai Poultry Company, Haimen, China), 140-day-old Gaoyou ducks (Jiangsu Gaoyou Duck Group, Yangzhou, China) and 70-day-old Yangzhou geese (Jiangsu Tian Ge Poultry Company, Hangzhou, China) were randomly selected, with 10 males and 10 females in each group. Before the experiment, the animals were pre-fed for 15 days with complete feed (Jiangsu Jinghai Poultry Company, Haimen, China) and water that was free of drugs and then fed for 2 weeks before slaughter. The selected animals were slaughtered on-site to reduce possible stress during transportation. All measures were taken to minimize the pain and suffering of the animals during the slaughter process. The bilateral chest muscles of each fowl were taken as the blank fowl muscle sample. Each sample was homogenized and fully mixed to obtain a blank sample, which was then packaged in a self-sealing bag at room temperature (25 °C) and stored at −34 °C for subsequent use.

Sample preparation involved extraction, purification, evaporation and redissolution. We subjected these muscle samples to liquid-liquid extraction (LLE) and SPE. Muscle samples (2.00 ± 0.02 g) were accurately weighed into a 50-mL stoppered centrifuge tube, 0.5 mL of 50% potassium hydroxide solution and 8 mL of ethyl acetate were added, and the mixture was vortexed for 5 min. Then, the mixture was sonicated for 5 min and centrifuged at 8000× *g* for 8 min at a temperature of 4 °C, and the supernatant was transferred to a 50-mL centrifuge tube. The extraction was repeated, and the supernatants were combined. The supernatants were transferred to a 100-mL separatory funnel, 8 mL of 0.1 mol/L hydrochloric acid solution was added and mixed, and the mixture was allowed to stand. The bottom layer was removed, and the extraction was repeated once. Extraction and purification were conducted on Oasis MCX (3 mL/60 mg, ELGA Lab Waters, High Wycombe, Bucks, UK) SPE cartridges that were sequentially equilibrated with 3 mL of methanol and 3 mL of 0.1 mol/L hydrochloric acid solution. After the extracts were loaded and filtered, the cartridges were washed successively with 3 mL of water and 3 mL of methanol. The pressure was reduced to 2.0 kPa, and the cartridges were allowed to dry for 3 min. The cartridges were then eluted with 3.0 mL of 5% ammoniated methanol. The eluates were collected in 10-mL centrifuge tubes and evaporated to dryness under a flow of nitrogen at 45 °C. The dried samples were resuspended in 2.0 mL of 0.1% aqueous formic acid and vortexed for 2 min. After passage through a 0.45-μm organic syringe filter, 10 μL of the filtrate was injected into the HPLC-MS/MS.

### 2.5. Method Validation

The proposed quantitative method for LMS, MBZ, HMBZ, and AMBZ detection in poultry muscle was validated by a series of quality parameters in accordance with recommendations from Commission Decision 2002/657/EC [25]. This legislation provides instructions on the performance of analytical methods and interpretation of results, and the validation of the methods used in this research is based on relevant content. With respect to sample verification, acceptable retention times of the four analytes to be tested, the signal-to-noise (S/N) ratio of each selected monitored ion, and the relative abundance ratio of the two monitored ions of each analyte in the sample were confirmed in accordance with Commission Decision 2002/657/EC [25] to meet the conditions for method validation.

The standard working solutions of LMS, MBZ, HMBZ, and AMBZ were diluted with appropriate amounts of different blank poultry muscle matrix extracts into six different series of concentrations, and the prepared mixed standard working solutions were analyzed by HPLC-MS/MS in sequence and five times at each concentration point to obtain the average value. The concentration of each analyte in the blank poultry muscle matrix sample was taken as the X-axis, the chromatographic peak areas of the quantitative ions (LMS, *m/z* 178.0/123.0; MBZ, *m/z* 264.0/104.8; HMBZ, *m/z* 265.8/160.0; AMBZ, *m/z* 105.0/76.9) were taken as the Y-axis, and the matrix standard curve was prepared as the quantitative curve of the sample to be tested.

The matrix effect (ME) refers to the influence of one or more coextracted compounds from the sample on the measurement of the analyte concentration or mass. We calculated the ME by comparing the slopes of the matrix-matched calibration curve to that of the solvent standard curve and evaluated ion suppression and enhancement.

After the extraction and purification of the blank sample, low concentrations of LMS, MBZ, HMBZ, and AMBZ standard working solutions were added, and the samples were analyzed by the optimized HPLC-MS/MS method. The concentration point of each standard solution was analyzed six times to calculate the average S/N ratio of confirmed product ions at the lowest quantitative level of each sample. When the S/N ratio of the product ions was greater than or equal to 3 (S/N ≥ 3), the added concentration of the corresponding drug was used as the limit of detection (LOD) of the analytical method. When S/N ≥ 10, the corresponding concentration was the limit of quantitation (LOQ) of the analytical method. Additionally, the LOQ concentration must meet the requirements of method establishment [23]. To demonstrate the reliability of the established analytical method, a recovery of no less than 70% and a relative standard deviation (RSD) of no more than 20% are generally required.

A blank sample (2.00 ± 0.02 g) was accurately weighed, and appropriate amounts of LMS, MBZ, HMBZ, and AMBZ standard working solutions were added. The LOQ, 0.5 MRL, 1.0 MRL, and 2.0 MRL solutions were analyzed in each blank, and six parallel experiments were performed at each level. The samples were analyzed by HPLC-MS/MS. The results obtained from the detection were substituted into the matrix standard curve of each analyte and quantified by the external standard method in the final step to calculate the concentration of each analyte in the sample. The ratio of the calculated concentration to the added concentration was the calculated recovery rate of the sample. Precision is usually expressed as RSD and is divided into intraday precision and interday precision. The same operator analyzed the above spiked samples using the same instrument and matrix standard curve at different times of the day and calculated the intraday RSD. The same operator analyzed the above spiked samples on different days of the week using a new matrix standard curve generated each day by the same instrument to calculate the interday RSD. Precision was determined based on the repeatability of the method and slight random fluctuation of the standard curve, instrument performance and environmental conditions.

CCα is defined as a limit value. When it is higher than the limit value, the probability error α can be used to draw the conclusion that a sample is unqualified. CCβ is defined as the lowest analyte concentration that can be detected, identified, and/or quantified in a sample with a β probability error. The MRL of LMS set by the European Union is 10 μg/kg, and the MRLs of MBZ, HMBZ, and AMBZ set by South Korea are 60 μg/kg. Therefore, in this study, CCα was calculated as MRL + 1.64 × standard deviation (α = 5%). CCβ was calculated as CCα+ 1.64 × standard deviation (β = 5%).

## 3. Results and Discussion

### 3.1. Optimization of HPLC-MS/MS Conditions

When developing HPLC-MS/MS methods, the mobile phase should be optimized first to improve the ionization efficiency of analytes and obtain an appropriate retention time to achieve good resolution and high sensitivity. In previous experiments, following the studies of Sun et al. [26] and Zhang et al. [27], we used 0.1% aqueous formic acid-methanol, 10 mM aqueous ammonium formate-methanol, 0.1% aqueous formic acid-acetonitrile, and 10 mM aqueous ammonium formate-acetonitrile as the mobile phases for optimization. When 10 mM ammonium formate was used, the baseline noise was excessive and fluctuating, while the baseline fluctuation was relatively stable when 0.1% aqueous formic acid was used. When the organic phase was methanol, the peak shape in the obtained chromatogram was bifurcated and exhibited tailing. The use of acetonitrile as the organic phase gave a good peak shape and higher recovery than that obtained with methanol. Therefore, in this experiment, 0.1% formic acid water and acetonitrile were used as the mobile phases for gradient elution. The retention time and peak area of the analytes were compared at column temperatures of 25 °C, 30 °C, 35 °C and 40 °C. When the column temperature was room temperature (25 °C), the retention time of the four analytes was 15 min, and the obtained peak areas were small. However, as the temperature increased, the chromatographic peak shapes of the four analytes improved. When the temperature of the chromatographic column was 35 °C, a larger peak area was obtained, and the drug peak extraction times were all within 10 min. The repeatability of the drug retention time was good, the pressure of the chromatographic column could be reduced, and the service life of the chromatographic column could be prolonged.

Optimal mass spectrometric conditions and appropriate ions were selected. Methanol and water (50:50 methanol/water, *v/v*) containing 0.1% formic acid were used to dilute each drug into a mass spectrometric tuning solution with a concentration of 50 ng/mL, which was directly injected into the mass spectrometer. The full-scan spectrum was obtained using ESI+, and the precursor ions were selected. In the experiment, the protonated molecular ion [M + H]^+^ of each compound was the most abundant ion. By optimizing the CE, two product ions were selected to determine the drug properties and conduct quantitative analysis. In the present method, the MRM mode was used for scanning and obtaining one precursor ion and two product ions for each compound. The MRM mode was selected because it has high sensitivity, good repeatability, and high precision. Two-stage ion selection eliminated a large number of interfering ions and significantly improved the S/N ratio of the target analyte to attain high detection sensitivity. In addition, separation of the ions to be tested and inhibition of mass spectral signals were avoided, which improved repeatability. The specificity of the MRM mode was used for continuous ion scanning analysis, and the obtained serial plain fragment data reduced the false-positive rate of qualitative results in the analysis process and ensured the accuracy of the analytical results. To ensure satisfactory peak shape and acceptable repeatability, 50 data points were collected for each peak using the automatic retention time setting. Optimizing the collection segment window for each analyte to reduce overlap significantly increased the residence time at each collection point and thus increased the intensity of the analyte. Finally, parameters, such as the ion source temperature, the ion source nozzle, the exit voltage of the collision chamber, the clustering voltage, and the CE were tested to screen the optimal parameters.

LMS, MBZ, HMBZ, and AMBZ are weakly basic drugs, and high sensitivity was obtained when the ESI+ mode was used. The samples were injected in the cross-flow mode via the needle pump, and mass spectrometric analysis was performed in the ESI+ mode. The first-stage mass spectrometric scans of LMS, MBZ, HMBZ, and AMBZ were performed within the scanning range of *m/z* 0–350 Da. The precursor ion information of the target compounds was obtained, which confirmed that the molecular weights of LMS, MBZ, HMBZ, and AMBZ were 205, 296, 298 and 238, respectively. Then, a product ion scan was used to obtain the product ion information of the target compounds, and the main fragment ions were mainly 178.0, 123.0 and 90.6 for LMS; 264.0, 104.8 and 149.0 for MBZ; 265.8, 160.0 and 78.9 for HMBZ; and 105.0, 76.9, and 132.8 for AMBZ. Figure 1 shows a two-stage mass spectrum formed after 50 superpositions of fragment ions generated from the collision of target compounds. Because the *m/z* of the precursor and product ions is the average of 50 scans, the error between the actual and theoretical *m/z* values for the precursor and product ions was within 0.1. The *m/z* values with the strongest ion abundance were selected as the monitoring ions: 178.0 and 123.0 for LMS, 264.0 and 104.8 for MBZ, 265.8 and 160.0 for HMBZ, and 105.0 and 76.9 for AMBZ. The MRM scanning mode was selected to optimize the mass spectral parameters.

LMS, MBZ, HMBZ, and AMBZ were characterized by *m/z* transitions of 205/178.0 and 205/123.0; 296/264.0 and 296/104.8; 298/265.8 and 298/160.0; and 238/105.0 and 238/76.9, respectively, and relative abundance ratios in this study. The analytes were quantified by the external standard curve method. The most abundant ions, *m/z* 205/178.0, 296/264.0, 298/265.8 and 238/105.0, were selected as the quantitative ions. During the determination of the spiked samples, the relative retention times of the target compounds LMS, MBZ, HMBZ, and AMBZ were all within 2.5% of the relative retention times in the corresponding external standard solutions, and the S/N ratio of each qualitative product ion was greater than or equal to 3. All of these analyses were performed using the same instrument conditions. The relative product ion abundance in the spiked samples was within the allowable range, as shown in Table 2.

### 3.2. Optimization of Sample Preparation

The sample pretreatment method should minimize interfering substances in the analytical solution to protect the instrument and column from damage and should be compatible with the analytical method used. In this experiment, LLE was combined with SPE to extract and purify LMS, MBZ, HMBZ, and AMBZ in the samples. LMS and MBZ and their metabolites have a certain protonation tendency under alkaline conditions [18], which can promote the extraction of the target substance from the tissue. When the extraction solution is alkaline, ionization of the analyte is inhibited, which reduces its solubility in the aqueous phase and increases its solubility in haloalkyl, ether, ester and other weakly polar organic solvents. Organic solvents, such as n-hexane and ethyl acetate can be used to extract the target analytes from animal tissue under alkaline conditions. Previous studies of these four analytes have primarily used acetonitrile [22] or ethyl acetate [28] as the extraction agent. To verify the optimal extraction process, the peak areas of each analyte in the poultry muscle matrix when acetonitrile and ethyl acetate were used as the extractants were compared. The analyte peak areas were largest when ethyl acetate was used as the extractant, so ethyl acetate was selected as the extractant for LMS, MBZ, HMBZ, and AMBZ in poultry meat in this study. Poultry samples were extracted under basic conditions with ethyl acetate and then back-extracted with hydrochloric acid.

When basic ethyl acetate was used as the extraction agent, a large number of lipid-like substances and other interfering substances in animal tissue samples were extracted into the organic phase, affecting the subsequent instrument determination. In addition, the concentration of the substance to be tested in the sample was relatively low, so purification, enrichment and concentration were required. When electrospray is used to ionize a drug in a complex matrix using an MS/MS system, sample preparation techniques, such as SPE are commonly employed to obtain a clean extract, reduce or avoid matrix effects, and reduce ion suppression [28]. Based on the selection of SPE cartridges and relevant studies, the extraction recoveries of Waters Oasis HLB cartridges, Waters Oasis MCX cartridges and Thermo Scientific HyperSep SCX cartridges were compared in this experiment. HLB cartridges contain an all-purpose hydrophilic and lipophilic balanced water-wettable reversed-phase adsorbent that maintains high retention and repeatability even after the filler has been pumped out. The MCX cartridge contains a mixed strong cation-exchange reversed-phase adsorbent with high selectivity for basic compounds and resistance to organic solvents. The SCX cartridge contains a strong cation exchanger and is mainly used to extract the strong cation exchange adsorption filler of charged basic compounds. The comparison showed that the recovery rate using MCX cartridges was the best. The MCX mixed-mode filter cartridge had hydrophobic and cation exchange retention properties. After purification by an MCX cartridge, the chromatographic peaks of the target drug and impurities could be separated well, the impurity peaks were significantly reduced, and the chromatographic peaks were well-shaped. In research on the analysis of veterinary drug residues, the MCX cartridge has often been used for the determination of weakly basic samples, such as plasma [29] and urine [30] with satisfactory results. Although the SCX cartridge was also suitable for extracting compounds under alkaline conditions, the presence of excessive fillers prolonged the time required for solid-phase extraction, and the recovery rate was not ideal. Therefore, the MCX cartridge was used for sample purification in this research.

### 3.3. Method Validation

#### 3.3.1. Specificity

The specificity of the established method was evaluated by detecting different blank poultry muscle samples and poultry muscle samples with different concentrations of the drug standard. Four concentrations (LOQ, 0.5 MRL, 1.0 MRL, and 2.0 MRL) of LMS, MBZ, HMBZ, and AMBZ were added to the blank poultry muscle (chicken, duck, goose). Figure 2 shows the total and extracted ion chromatograms of a sample with the LOQ of the standard added.

#### 3.3.2. Linearity

In this study, linearity was estimated by matrix-matched calibration standard curves. The concentrations of the four analytes in the blank chicken muscle matrix samples were as follows: LMS, 0.22–25 μg/kg; MBZ, 0.15–150 μg/kg; HMBZ, 0.60–150 μg/kg; and AMBZ, 0.80–150 μg/kg. The concentrations of the four analytes in the blank duck muscle matrix samples were as follows: LMS, 0.20–25 μg/kg; MBZ, 0.12–150 μg/kg; HMBZ, 0.50–150 μg/kg; and AMBZ, 0.62–150 μg/kg. The concentrations of the four analytes in muscle matrix samples of different blank geese were as follows: LMS, 0.16–25 μg/kg; MBZ, 0.16–150 μg/kg; HMBZ, 0.55–150 μg/kg; and AMBZ, 0.70–150 μg/kg. There was a good linear relationship between the peak area (Y) of the quantitative product ions of the four compounds and their added concentration (X), and R^2^ values were higher than 0.9990. The linear regression equations, measurement coefficients and linear ranges of LMS, MBZ, HMBZ, and AMBZ in poultry muscle (chicken, duck, goose) are shown in Table 3. If the concentration of an analyte exceeded the linear range of the sample, the sample was diluted so that the concentration of the analyte was within this range, and the obtained concentration was multiplied by the dilution factor to obtain the concentration of the original sample.

#### 3.3.3. Matrix Effect

The MEs were calculated according to the following equation:ME (%) = [(Slope _matrix-matched calibration curve_/Slope _solvent standard curve_) − 1] × 100%

An ME value between −20% and 20% is considered to reflect an acceptable weak ME; an ME value of −50% to −20% or 20% to 50% is medium, and an ME below −50% or above 50% is considered to be strong. The signal is enhanced if the value is positive and suppressed if the value is negative [31]. As shown in Table 4, slight ion enhancement or ion attenuation of the target compound occurred in all three poultry matrices, but all matrix effects were within the acceptable range (−13.6 to 17.3). The results showed that the ME was effectively compensated in this study by using a matrix-matched calibration curve and solvent standard curve, and similar results were obtained by Yoshikawa et al. [32] in a study of a chicken matrix.

#### 3.3.4. LODs and LOQs

In general, relatively low LODs and LOQs indicate the high sensitivity of the detection method. In this study, HPLC–MS/MS was used, and the MRM mode was adopted to detect the four target analytes in poultry muscle and improve the sensitivity of the detection method. As shown in Table 5, the LODs of LMS, MBZ, HMBZ, and AMBZ in poultry muscle were 0.05–0.07, 0.04–0.06, 0.15–0.18, and 0.23–0.30 μg/kg, and the LOQs were 0.16–0.22, 0.12–0.16, and 0.50–0.80 μg/kg, respectively, under the conditions of this study. Compared with other HPLC-MS/MS [32] and UPLC-MS/MS methods [22], the detection method obtained herein showed lower limits and better sensitivity than others reported in the literature.

#### 3.3.5. CCα and CCβ

CCα and CCβ were calculated by analyzing a standard solution of 60 blank muscle matrixes (20 for each chicken, duck and goose) at the level of MRL (LMS: 10 μg/kg; MBZ, HMBZ, AMBZ: 60 μg/kg), and the relevant data are listed in Table 5. The values of CCα and CCβ calculated in this study were similar to the MRL values.

#### 3.3.6. Recovery and Precision

The recovery represents the retention of the target analyte after sample pretreatment and can be used to measure the reliability of the established analysis method. Precision can reflect errors throughout the study. Intraday RSDs and interday RSDs can be used to determine the precision of the method. Samples containing the analytes at four concentrations (LOQ, 0.5 MRL, 1.0 MRL, and 2.0 MRL) were analyzed at different time points on the same day with the same instrument and the same standard curve. Each added concentration was analyzed six times, and the intraday (intrabatch) precision was calculated. The purpose of determining the precision was to test the reproducibility of the method. Samples containing the analytes at four different concentrations (LOQ, 0.5 MRL, 1.0 MRL, and 2.0 MRL) were measured on different days of a week with new standard curves generated every day on the same instrument. Each concentration was analyzed six times, and the interday (interbatch) precision was calculated. This analysis was conducted to investigate the effects of the standard curve, instrument performance, environmental conditions and other small random fluctuations on the analysis. The smaller the intraday precision and RSDs, the more reliable the method. The recoveries of LMS, MBZ, HMBZ, and AMBZ in the blank poultry (chicken, duck, and goose) muscles at the four spiked concentrations (LOQ, 0.5 MRL, 1.0 MRL, and 2.0 MRL) are presented in Table 6. The results showed that the method was stable and reliable and fully met the requirements of residual analytical methods for the detection of LMS, MBZ, HMBZ, and AMBZ in poultry muscle.

### 3.4. Comparison with Other Methods

Many methods for detecting LMS, MBZ and its two metabolites in animal-derived foods have been reported, and these methods were investigated and compared with the sample preparation method and detection method. A comparison of the test results for the analysis of the target analyte using the different methods is presented in Table 7. Table 7 shows that the sensitivity of ultraviolet detection and fluorescence detection is relatively low, while the obtained detection limit and quantitation limit are high. When MS is used, higher sensitivity and accuracy as well as a lower detection limit and quantitation limit can be obtained. In recent studies, Xu et al. [21] developed an LC-MS/MS method for the analysis of LMS, MBZ, HMBZ, and AMBZ residues in aquatic products with good results. Sample preparation was performed by an improved QuEChERS method using a Poroshell 120 EC C_18_ column. The LODs and LOQs for all analytes tested were below 0.3 μg/kg and 1 μg/kg, respectively. The recovery of the analytes was above 80.0%. Zhu et al. [33] developed an LC-MS/MS analytical method for the simultaneous determination of 88 commonly used veterinary drugs, including LMS, MBZ, HMBZ, and AMBZ, in milk. The LODs and LOQs were in the ranges of 0.2–2.0 μg/kg and 0.5–10 μg/kg, respectively. The mean recovery of the spiked target compound ranged from 63.1% to 117.4% at various levels of addition. In this study, LLE combined with SPE was used for sample preparation, and HPLC-MS/MS was used for poultry muscle sample analysis. Compared with other methods, the developed method had better LOD, sensitivity and recovery.

### 3.5. Real Sample Analysis

To evaluate the reliability, suitability and feasibility of the newly developed method, 60 poultry samples (20 chicken, 20 duck and 20 goose samples) were purchased from a supermarket in our city, and all samples were tested for residues using the optimized method. LMS was detected at 4.06 µg/kg and 3.48 µg/kg in two chicken samples, and MBZ was detected at 9.82 µg/kg in one goose sample. None of the samples contained drug residues exceeding the MRLs of the EU (10 µg/kg) and South Korea (60 µg/kg). LMS, MBZ, AMBZ, and HMBZ residues were not detected in any of the other samples. Therefore, the HPLC-MS/MS method established in this study can be used as a reliable method for the detection of LMS, MBZ and its two metabolite residues.

## 4. Conclusions

In the present study, a rugged and effective HPLC-MS/MS method was developed for the determination of LMS, MBZ and its two metabolites in poultry muscles. Using external standard quantitative analysis, the recovery rates of the four analytes were above 86.00%, and the LODs were below 0.30 μg/kg, fulfilling the requirements for the simultaneous detection of the four target analytes in poultry muscle. The method has the advantages of low matrix interference, high sensitivity and recovery rate, and strong anti-interference capability and fulfills the requirements in the confirmatory criteria of the European Commission. Additionally, 60 real samples were analyzed by this method, validating the reliability, applicability and feasibility of the established method. The study provides technical support for the simultaneous detection of LMS and MBZ and their two metabolites in poultry muscle.

## Figures and Tables

**Figure 1 foods-10-02841-f001:**
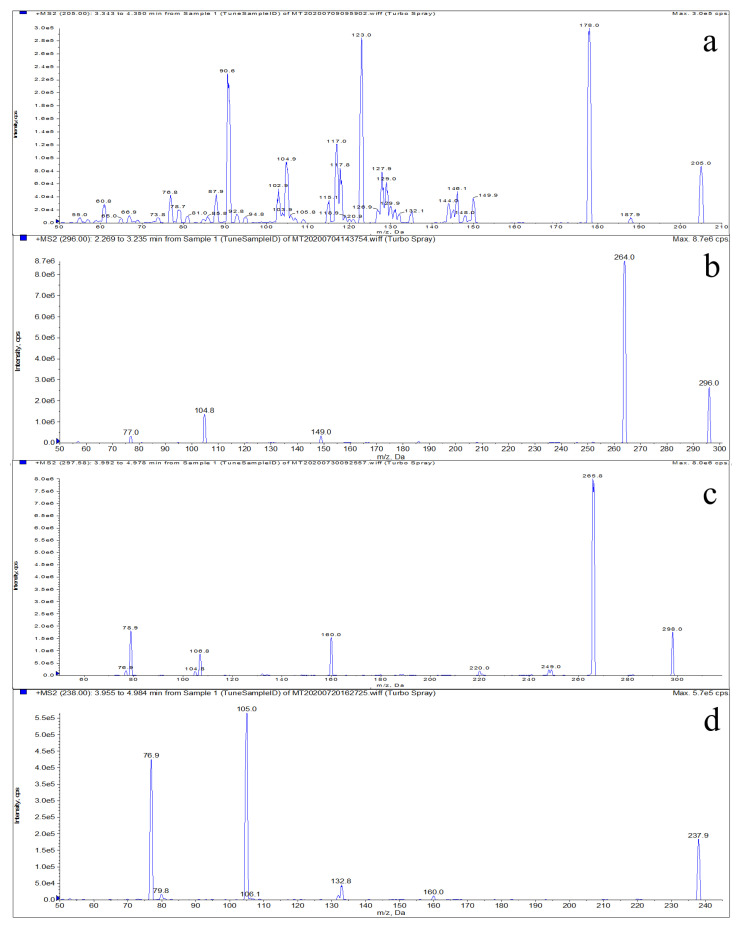
Mass spectra of LMS (**a**), MBZ (**b**), HMBZ (**c**) and AMBZ (**d**).

**Figure 2 foods-10-02841-f002:**
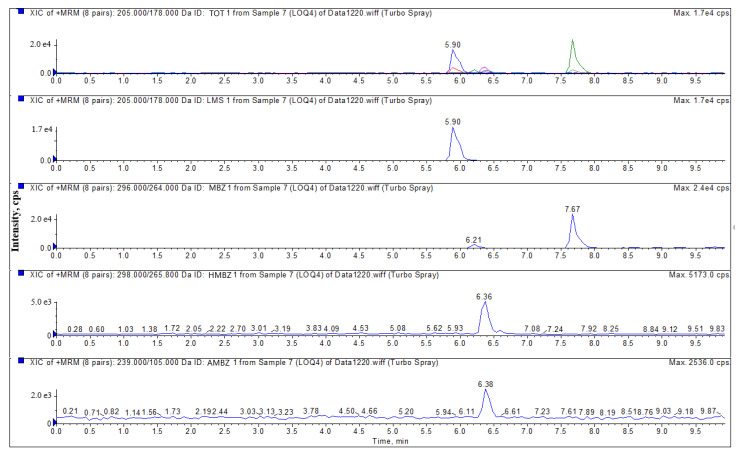
Total ion chromatograms and extracted ion chromatograms of a blank duck muscle matrix spiked with the LOQ of the standard added. The colored peaks on the total ion chromatogram correspond to those of the four analytes: LMS, blue; MBZ, green; HMBZ, rose red; AMBZ, light blue.

**Table 1 foods-10-02841-t001:** HPLC-MS/MS conditions and retention times for the analysis of LMS, MBZ, HMBZ and AMBZ.

Compound	Molecular Weight	Retention Time (min)	Mass Transition (*m/z*)	Declustering Potential (V)	Collision Energy (eV)
LMS	205	5.91	205 > 178.0 *	110	29
			205 > 123.0		38
MBZ	296	7.68	296 > 264.0 *	115	28
			296 > 104.8		23
HMBZ	298	6.37	298 > 265.8 *	121	24
			298 > 160.0		35
AMBZ	238	6.38	238 > 105.0 *	155	33
			238 > 76.9		45

Note: LMS, levamisole; MBZ, mebendazole; HMBZ, 5-hydroxymebendazole; AMBZ, 2-amino-5-benzoylbenzimidazole; *, quantificational ion pair.

**Table 2 foods-10-02841-t002:** Ion ratios of two transition reactions of the four analytes in standard solutions and spiked samples.

Analyte	Ion Ratio of Standard Solutions	Maximum Permitted Tolerance According to Decision 2002/657/EC	Ion Ratio of Fortified Samples
LMS	0.26	0.26 ± 25% (0.11–0.51)	0.22–0.36
MBZ	0.20	0.20 ± 25% (0.10–0.45)	0.13–0.25
HMBZ	0.98	0.98 ± 20% (0.78–1.18)	0.78–1.11
AMBZ	0.62	0.62 ± 20% (0.42–0.82)	0.50–0.74

**Table 3 foods-10-02841-t003:** The linear regression equation, determination coefficient and linearity range of LMS, MBZ, HMBZ and AMBZ in poultry muscle.

Matrix	Analyte	Regression Equation	Determination Coefficient (R^2^)	Linearity Range (μg/kg)
Chicken muscle	LMS	y = 891,231x − 13,830	0.9995	0.22–25
	MBZ	y = 298,982x + 168,358	0.9996	0.15–150
	HMBZ	y = 89,829x − 4855	0.9995	0.60–150
	AMBZ	y = 31,616x − 4274	0.9996	0.80–150
Duck muscle	LMS	y = 496,991x + 336,140	0.9995	0.20–25
	MBZ	y = 207,883x + 79,368	0.9994	0.12–150
	HMBZ	y = 37,850x + 4580	0.9994	0.50–150
	AMBZ	y = 22,019x + 1032	0.9995	0.62–150
Goose muscle	LMS	y = 858,472x + 38,029	0.9998	0.16–25
	MBZ	y = 313,426x + 148,749	0.9997	0.16–150
	HMBZ	y = 83,271x − 12,277	0.9995	0.55–150
	AMBZ	y = 30,999x − 1204	0.9997	0.70–150

**Table 4 foods-10-02841-t004:** MEs of LMS, MBZ, HMBZ and AMBZ in poultry muscle (%).

Analyte	Chicken Muscle	Duck Muscle	Goose Muscle
LMS	−7.2	−5.2	−13.6
MBZ	4.9	12.9	9.2
HMBZ	3.6	−8.8	−10.6
AMBZ	7.8	13.1	17.3

**Table 5 foods-10-02841-t005:** LOD, LOQ, CCα and CCβ of LMS, MBZ, HMBZ and AMBZ in poultry muscle.

Matrix	Analyte	LOD (µg/kg)	LOQ (µg/kg)	CCα (µg/kg)	CCβ (µg/kg)
Chicken muscle	LMS	0.07	0.22	14.58	19.15
	MBZ	0.06	0.15	63.33	66.66
	HMBZ	0.16	0.60	63.46	66.92
	AMBZ	0.25	0.80	63.80	63.60
Duck muscle	LMS	0.06	0.20	13.25	16.49
	MBZ	0.04	0.12	63.69	67.38
	HMBZ	0.15	0.50	63.15	66.30
	AMBZ	0.30	0.62	63.41	66.82
Goose muscle	LMS	0.05	0.16	16.45	22.90
	MBZ	0.05	0.16	64.30	68.60
	HMBZ	0.18	0.55	56.18	70.36
	AMBZ	0.23	0.70	67.35	74.70

**Table 6 foods-10-02841-t006:** Recovery and precision of LMS, MBZ, HMBZ and AMBZ added to blank poultry muscle (*n* = 6).

Matrix	Analyte	Addition Level (µg/kg)	Recovery (%)	RSD (%)	Intraday RSD (%)	Interday RSD (%)
Chicken muscle	LMS	0.22	88.01 ± 2.29	2.29	3.51	3.83
		5	91.44 ± 1.93	2.11	2.07	4.03
		10 ^α^	93.30 ± 2.79	2.99	3.68	4.34
		20	93.89 ± 3.22	3.43	3.30	4.24
	MBZ	0.15	86.77 ± 2.24	2.58	3.51	3.98
		30	94.05 ± 1.96	2.09	2.07	2.62
		60 ^α^	94.38 ± 2.03	2.15	3.68	3.99
		120	94.49 ± 2.18	2.31	3.30	3.73
	HMBZ	0.60	88.36 ± 3.65	4.13	3.21	4.60
		30	91.42 ± 2.01	2.20	4.56	4.69
		60 ^α^	93.21 ± 2.11	2.26	2.39	2.96
		120	91.53 ± 2.53	2.77	4.55	5.07
	AMBZ	0.80	90.70 ± 1.90	2.09	2.60	3.06
		30	92.07 ± 1.95	2.11	2.49	2.95
		60 ^α^	94.49 ± 2.32	2.45	3.92	4.25
		120	96.47 ± 4.46	4.63	3.25	4.96
Duck muscle	LMS	0.20	90.85 ± 2.95	3.25	3.17	4.06
		5	89.57 ± 2.05	2.29	3.56	3.89
		10 ^α^	92.04 ± 1.98	2.16	2.52	2.99
		20	95.95 ± 2.67	2.78	5.13	5.37
	MBZ	0.12	88.68 ± 2.83	3.19	3.85	4.51
		30	91.08 ± 1.94	2.13	2.46	2.95
		60 ^α^	95.14 ± 2.25	2.36	2.13	2.87
		120	95.21 ± 1.97	2.07	3.52	3.77
	HMBZ	0.50	91.33 ± 2.62	2.86	4.99	5.28
		30	95.63 ± 3.71	3.88	2.47	4.03
		60 ^α^	92.26 ± 1.92	2.08	4.03	4.18
		120	93.86 ± 2.06	2.20	2.52	3.03
	AMBZ	0.62	89.00 ± 2.23	2.51	3.19	3.69
		30	89.95 ± 3.32	3.69	3.64	4.63
		60 ^α^	95.58 ± 2.08	2.17	2.68	3.12
		120	96.94 ± 2.93	3.02	3.04	3.85
Goose muscle	LMS	0.16	87.58 ± 2.21	2.52	4.36	4.62
		5	90.54 ± 4.07	5.19	3.01	5.22
		10 ^α^	95.03 ± 3.93	4.14	2.34	4.12
		20	95.88 ± 2.81	2.93	3.73	4.29
	MBZ	0.16	87.62 ± 1.87	2.13	4.67	4.75
		30	90.05 ± 2.31	2.57	2.52	3.23
		60 ^α^	95.18 ± 2.62	2.75	3.55	4.12
		120	93.68 ± 2.97	3.17	2.92	3.83
	HMBZ	0.55	90.25 ± 2.12	2.35	3.34	3.73
		30	93.28 ± 4.03	4.32	4.33	5.52
		60 ^α^	94.79 ± 3.16	3.33	3.23	4.16
		120	92.97 ± 2.92	3.14	3.15	4.03
	AMBZ	0.70	92.71 ± 3.31	3.57	3.73	4.66
		30	92.84 ± 2.13	2.29	1.75	2.54
		60 ^α^	96.00 ± 4.48	4.67	2.08	4.39
		120	95.49 ± 2.96	3.10	3.04	3.91

Note: ^α^, maximum residue limit.

**Table 7 foods-10-02841-t007:** Comparison of detection results between this research method and other LC methods.

Matrix	Analyte	Chromatographic Conditions	Sample Preparation Method	Recovery (%)	LODs (μg/kg)	LOQs (μg/kg)
Chicken muscle [32]	37 veterinary drugs (including LMS, MBZ, HMBZ)	Inert Sustain Swift C_18_ column (250 mm × 4.6 mm, 5 μm) Mobile phase: 0.1% formic acid in water and 10 mM ammonium acetate in methanol HPLC-MS/MS	Extract with ethyl acetate solution and acetonitrile	76–84	0.4	-
Milk [14]	21 veterinary drugs (including MBZ, HMBZ, AMBZ)	Waters XTerra C_18_ column (250 × 4.6 mm, 5 μm) Mobile phase: 0.02 M ammonium acetate and acetonitrile HPLC-UVD	Extract with acetonitrile; Purification of MCX solid phase extraction column	78–109	3	10
Beef, pork and poultry muscle [28]	LMS	ODS-80Ts column (4.6 mm × 150 mm, 5 μm) Mobile phase: 0.02 mol/L aqueous potassium phosphate solution and acetonitrile HPLC-DAD	Extract with ethyl acetate solution; Purification of SCX solid phase extraction column	78.30–99.80	5	-
Pork, chicken, horse muscle [26]	MBZ, HMBZ, AMBZ	Unison UK C_18_ column (100 mm × 2 mm, 3 μm) Mobile phase: 10 mM ammonium formate in water and methanol HPLC-MS/MS	Extract with ethyl acetate solution	86.30–101.28	0.07	0.2
Milk [33]	88 veterinary drugs (including 13 benzimidazoles)	PAK C_18_ MG column (150 mm × 2.1 mm, 5 μm) Mobile phase: acetonitrile and 0.1% formic acid aqueous solution HPLC-MS/MS	Extract with disodium ethylenediaminetetraacetate and acetonitrile solution	75.5–104.5	0.5	2.0
Aquatic product [21]	LMS, MBZ, HMBZ, AMBZ	Poroshell 120 EC C_18_ column (250 mm × 4.6 mm, 5 μm) Mobile phase: 5 mM ammonium formate in methanol and acetonitrile LC-MS/MS	Extract with acetonitrile	80–113.7	0.3	1
Beef [22]	37 veterinary drugs (including LMS and MBZ)	Hss T3 C_18_ column (100 mm × 2.1 mm, 1.8 μm) Mobile phase: acetonitrile:0.01% formic acid in water (10:90, *v/v*) and methanol:5 mM ammonium formate UHPLC-MS/MS	Extract with acetonitrile	81–101	0.13–1.55	0.22–2.64
Milk [34]	19 veterinary drugs (including MBZ, HMBZ, AMBZ)	Ascentis Express C_18_ column (150 mm × 2.1 mm, 2.7 μm) Mobile phase: 0.1% formic acid and acetonitrile UHPLC-MS/MS	Extract with 10 mL of acetonitrile with 0.1% NH_3_ followed by 5 g of MgSO4:NaCl (4:1, *w/w*)	65–100	1.0–10	1.6–18
Fish [35]	71 veterinary drugs (including LMS and MBZ)	X-SELECT HSS C_18_ (150 mm × 2.1 mm, 3.5 μm) Mobile phase: 0.1% formic acid in 2 mM ammonium formate in water and 0.1% formic acid in acetonitrile LC-MS/MS	Extract with 0.1% formic acid in MeCN/MeOH (95:5, *v/v*) clean up with PSA and C_18_ d-SPE sorbent	60–119	-	0.02–4.8
Animal-derived food [36]	176 veterinary drugs (including LMS and MBZ)	Hss T3 C_18_ column (100 mm × 2.1 mm, 1.8 μm) Mobile phase: water with 0.1% formic acid and acetonitrile/MeOH (50:50, *v/v*) with 0.1% HCO_2_H UHPLC-MS/MS	Extract with acetonitrile/water (4:1, *v/v*) Clean up with EMR-Lipid cartridge	70–120	-	-
Chicken, duck, goose muscle (The study)	LMS, MBZ, HMBZ, AMBZ	Xbridge^TM^ C_18_ column (4.6 mm × 150 mm, 5 μm) Mobile phase: 0.1% formic acid in water and acetonitrile HPLC-MS/MS	Extract with ethyl acetate; Purification of MCX solid phase extraction column	86.77–96.94	0.04–0.30	0.12–0.80

## Data Availability

All available data are contained within the article.

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
