# Peer review of "Determination of Levamisole and Mebendazole and Its Two Metabolite Residues in Three Poultry Species by HPLC-MS/MS"

_foods, 2021, doi:10.3390/foods10112841_

Round 1

Reviewer 1 Report

The submitted paper deals with development of a high-performance liquid chromatography-tandem mass spectrometry (HPLC-MS/MS) method for simultaneous analysis of levamisole (LMS) and mebendazole (MBZ) and its two metabolites; 5-hydroxymebendazole (HMBZ) and 2-amino-5-benzoylbenzimidazole 17 (AMBZ), in poultry muscle (chicken, duck and goose).

The manuscript is scientifically sound. The abstract is informative and presents an overview of the whole paper. The introduction provides the reader with an insight into the subject in question. The method development and validation is appropriately presented and supplemented with the respective tables and figures. The conclusions are consistent with the objectives of the manuscript.

This study provides further evidence supporting the use of this technique for simultaneous determination of respective endoparasites in poultry muscles. However, some clarification or additional information and data is needed before it is considered for publication. 

The authors should address the following points:

Introduction

  • Add authors who have published simultaneous detection of LMS and MBZ in other food matrices (e.g. https://doi.org/10.1271/bbb.70.54; https://doi.org/10.1016/j.chroma.2009.07.036)
  • The last paragraph should not contain the conclusions of the work, only the aim of the study.
  • Some terms and expressions presented in the manuscript are incorrect (e.g. some countries, such as the European Union [7], the United States [8] and South Korea; animal-origin foods).

Material and Methods

  • In the first two paragraphs, there are written two different temperatures for storage of standard stock solution in a freezer (-34 °C and -72°C, please clarify).
  • Preparation of the sample – The experimental treatment design is totally missing (drugs, feed, brand names, manufacturers, route of administration, doses, animal slaughter /welfare rules?/, the place where the experiment was carried out…).
  • Method validation – First paragraph: Did you mean Commission Decision 2002/657/EC?; the legislation should be specified at its first occurrence in the text as both references are linked to the same legislation.

Results and Discussion

  • Optimization of HPLC-MS/MS, conditions and sample preparation are described in great details. I positively evaluate the part “Comparison with other methods” and using the real samples for evaluation of the suitability of this method for the detection of LMS, MBZ and its two metabolite residues.

Conclusions

  • The abbreviation of the method (HPLC-MS/MS, LLE-SPE-HPLC–MS/MS) method should be given uniformly.

Author Response

Thank you for this affirmation. We have revised the manuscript in accordance with your comments, and our point-by-point responses follow. We hope that our manuscript will now meet the journal’s requirements.

Reviewer 2 Report

The article is well written, well structured, and it is perceived that the authors struggled to perform validation in each muscle matrix (chicken, duck and goose).

I recommend, though, in order to improve the paper, minor suggestions:

Lines 2-3: I suggest to replace the title by: Determination of Levamisole and Mebendazole and Its Two Metabolite Residues in three Poultry species by HPLC-MS/MS

Line 31: As the title already has most of the key-words cited by the authors, the readers will find the manuscript already with the mentioned key-words. Therefore, I suggest the new-keywords to be:

Chicken meat, goose meat, duck meat, veterinary drugs, validation method

Line 118: was it necessary to buy ammonia? Or only ammonium formate?

Line 193: prefed? Would’t it be fasted (meaning that animals were deprived from food)

Line 198: at -34ºC

Lines 441-442: rewrite sentence to be clear: the detection method obtained herein showed lower limits and better sensitivity than others reported in the literature (Table 6)

Line 470: ... Table 6, which shows ...

Line 485:  .. LOD, sensitivity and recovery.

Line 514: Funding (is repeated twice)

Author Response

Thank you for reviewing our manuscript and for affirming our work. Your professional suggestions will have a positive impact on our future work. We have revised the manuscript according to your comments, and the changes are marked in red in the manuscript.

Reviewer 3 Report

The manuscript is well written, but some improvements should be implemented and some points should be clarified and commented.

Please see the following:

1) In the introduction, the differences in the regulation in EU and South Korea concerning MRL for mebendazol should be clarified (the species for which an MRL has been assigned in EU should be listed) especially that MRL in EU has been not established for poultry

2) Sample preparation in presented method is much complicated and time-consuming, (extraction steps repeated many times with SPE) like for methods with UV or FLD detectors. This is evident even in the Table 6 where comparison with others LC methods is presented. In most of LC-MS/MS methods only extraction with solvents are implemented, while for UVD or DAD extraction with solvent and purification with SPE columns are described. Such methods, including method presented in this paper, are not practical in the laboratory work focused on the control of veterinary drug residues in food of animal origin, which is related with lack of novelty in the method development.

3) Why authors not use internals standards in presented LC-MS/MS method? 

4) Matrix effect should be estimated and calculated.

5) In Table 1 CXP and dwell time should be inserted.

6) Too many, unnecessary ion chromatograms. Why authors present chromatograms for LMS at 1 ug/kg and for MBZ, HMBZ and AMBS at 6 ug/kg. These are relatively high concentrations in comparison with established LOD and LOQ at 0.04 – 0.3 ug/kg and 0.12 – 0.8 ug/kg. Chromatograms at LOQ level should be presented. 

7) Authors stated that method was validated according to 2002/657/EC. Why decision limit  CCα one of the most important parameters in residues control as well as  CCβ are not estimated? According to which regulation LOD and LOQ were calculated, because in 2002/657/EC these parameters are not included.

8) The references should be update, there are some more up-to-date methods for determination compounds of interest.

Author Response

Thank you for your meticulous review and professional suggestions and opinions. We have made detailed revisions of the manuscript according to your suggestions, and our point-by-point responses to your comments follow. We hope that the cover letter and the revised manuscript meet your requirements. 

Round 2

Reviewer 3 Report

1) Authors mentioned that complicated and time-consuming sample preparation and SPE purification was applied to remove and purify impurities as much as possible. So the more matrix effect (ME %) should be calculated and these data should be completed.

2) Authors agree that this sample preparation and purification is not an innovative, so what's the point of publishing such a work? Authors stated that in subsequent work, they will apply new sample preparation technologies, such as accelerated solvent extraction and QuEChERS, to improve the efficiency and practicality of laboratory work. So that work should be described and published.

3) Still lack of internal standards, which are available and necessary in LC-MS/MS method, despite automatic sample injection.

4) With such good purification in the presented method as authors stated, the peaks of impurities at chromatograms should be reduced, even at LOQ levels.

Authors stated: “when the concentrations were equivalent to the LOQ and LOD, the response values of the target compounds were relatively low, and there were many chromatographic peaks of impurities, which reduced the quality of the ion chromatograms. When LMS was administered at 1 μg/kg and MBZ, HMBZ and AMBZ were administered at 6 μg/kg, the impurity chromatographic peaks were greatly reduced.”

But 0.04 – 0.3 μg/kg and 0.12 – 0.8 μg/kg as LOD and LOQ are far away from presented concentrations as 1 and 6 ug/kg. Maybe the method should be more refined or the LOD, LOQ values raised?

Author Response

Dear Editor and Referee:

We greatly appreciate the efficient, professional and rapid processing of our contribution (Manuscript ID: foods-1427640) by editor, and we also appreciate the meticulous work of the assigned editor in reviewing our previous manuscript.

We modified the manuscript with point-by-point revisions based on the suggestions and comments of the referee, and all revisions in the current version are highlighted in red. We hope that the quality and format of the manuscript meet the requirements of the journal.

The specific revisions are as follows.

  • Authors mentioned that complicated and time-consuming sample preparation and SPE purification was applied to remove and purify impurities as much as possible. So the more matrix effect (ME %) should be calculated and these data should be completed.

Answer: Thank you for your professional advice. We carried out a complementary experiment to calculate the matrix effect. The calculated MEs were added to the manuscript, and all changes have been marked in red. Please refer to lines 240-243 and 431-443 of the manuscript.

  • Authors agree that this sample preparation and purification is not an innovative, so what's the point of publishing such a work? Authors stated that in subsequent work, they will apply new sample preparation technologies, such as accelerated solvent extraction and QuEChERS, to improve the efficiency and practicality of laboratory work. So that work should be described and published.

Answer: Thank you for your comments. The application of QuEChERS and ASE methods will be addressed in future work, so they cannot be discussed or made public at this time. Although QuEChERS and ASE purification methods are simpler and more convenient, since not all laboratories are equipped with ASE and the cost of QuEChERS is high, the traditional sample processing method we applied is convenient for more widespread laboratory use. By comparing the test results of different extraction methods and purification methods, the appropriate sample extraction agent and purification step were selected. Compared with other related studies, the optimized method obtained a better recovery rate and lower detection limit. We optimized all the procedures for better results. I hope you agree with our methods, and we thank you very much for your professional evaluation.

  • Still lack of internal standards, which are available and necessary in LC-MS/MS method, despite automatic sample injection.

Answer: Thank you for your suggestions. I am very sorry for causing confusion. As the manuscript shows, we have completed this experiment. However, because no suitable internal standard substance was found in the course of the study, another common method, the external standard method, was chosen for the calculation of the recovery. We ensured the stability and reproducibility of the assay in our work by precise autoinjection, ensured the consistency of the experimental conditions during operation to improve the recovery, and finally obtained relatively satisfactory results. We hope that you understand our approach, and we hope that our responses will be satisfactory. If you feel that additional responses are required, we hope that you will give us another opportunity to contact you.

  • With such good purification in the presented method as authors stated, the peaks of impurities at chromatograms should be reduced, even at LOQ levels.

Authors stated: “when the concentrations were equivalent to the LOQ and LOD, the response values of the target compounds were relatively low, and there were many chromatographic peaks of impurities, which reduced the quality of the ion chromatograms. When LMS was administered at 1 μg/kg and MBZ, HMBZ and AMBZ were administered at 6 μg/kg, the impurity chromatographic peaks were greatly reduced.”

But 0.04 – 0.3 μg/kg and 0.12 – 0.8 μg/kg as LOD and LOQ are far away from presented concentrations as 1 and 6 ug/kg. Maybe the method should be more refined or the LOD, LOQ values raised?

Answer: Thank you for the comments. Our previous response may have been unclear, and we apologize for the inconvenience. As you said, chromatograms at the LOQ level should be presented. We have added chromatograms at the LOQ level to the manuscript as you recommended; see lines 410-412 for details. We hope our responses you are satisfactory, and we thank you for your professional review work.
